# Identification of Small Peptides that Inhibit NADPH Oxidase (Nox2) Activation

**DOI:** 10.3390/antiox7120181

**Published:** 2018-12-05

**Authors:** Aron B. Fisher, Chandra Dodia, Sheldon I. Feinstein

**Affiliations:** Institute for Environmental Medicine, University of Pennsylvania Perelman School of Medicine, Philadelphia, PA 19104, USA; cdodia@pennmedicine.upenn.edu (C.D.); sif@pennmedicine.upenn.edu (S.I.F.)

**Keywords:** peroxiredoxin 6, surfactant protein A, phospholipase A_2_, drug delivery, liposomes

## Abstract

Nicotinamide adenine phosphate (NADPH) oxidase type 2 (Nox2), a major source of reactive oxygen species in lungs, plays an important role in tissue damage associated with acute inflammatory diseases. The phospholipase A_2_ (PLA_2_) activity of peroxiredoxin 6 (Prdx6), called aiPLA_2_, is required for Nox2 activation through its role in the cellular generation of Rac, a key cytosolic component of the activation cascade. Lung surfactant protein A (SP-A) binds to Prdx6, inhibits its aiPLA_2_ activity, and prevents activation of Nox2. Based on protein docking software, we previously identified a 16 amino acid (aa) peptide derived from rat SP-A as the Prdx6 binding motif. We now identify the minimal effective sequences of rat/mouse and human SP-A as 9-aa sequences that we have called PLA_2_-inhibitory peptide (PIP).These sequences are PIP-1, rat/mouse; PIP-2, human; and PIP-3, a hybrid of PIPs 1&2. aiPLA_2_ activity in vitro was inhibited by 50% with ~7–10 µg PIP/µg Prdx6. Inhibition of the aiPLA_2_ activity and Nox2 activation of lungs in vivo was similar for intratracheal (IT) and intravenous (IV) administration of PIP-2, but required its incorporation into liposomes as a delivery vehicle; tissue ½ time for decrease of the in vivo inhibition of aiPLA_2_ activity after PIP-2 administration was ~50 h. These properties suggest that PIP-2 could be an effective therapeutic agent to prevent tissue injury associated with lung inflammation.

## 1. Introduction

The term ‘reactive oxygen species’, abbreviated as ROS, comprises superoxide anion (O_2_**^−•^**), a free radical, and the non-radical H_2_O_2_, along with other chemically reactive compounds derived from the metabolism of molecular O_2_. ROS are generated normally during cellular metabolism by a variety of enzymatic as well as non-enzymatic pathways. However, the seven-member family of NADPH oxidases (Nox) and related dual oxidases (Duox) are the only enzymes that are known to generate ROS as their primary product [1]. Nox2, the first of the NOX/Duox family of enzymes to be described, is expressed widely in tissues with especially high levels in polymorphonuclear leukocytes (PMN) and macrophages, where it is localized primarily to intracellular phagolysosomes, and in pulmonary and cardiovascular endothelium where it is localized to the plasma membrane [2]. Intracellular Nox2 is inactive in the resting state but can be activated by a complex and tightly controlled pathway [3]. Control of activation is critical since the uncontrolled production of ROS can lead to oxidative damage to tissues.

The activation of the enzymatic activity of Nox2 to produce ROS requires the phosphorylation of several cytosolic proteins (p67^phox^, p47^phox^, and p40^phox^) and the activation of Rac (1 or 2, depending on the cell type) [4,5,6]. Activation of Rac occurs through its dissociation from a complex with Rho protein GDP dissociation inhibitor (Rho-GDI) and formation of Rac-GTP. Activated Rac binds to phosphorylated p67^phox^, facilitating its translocation (along with other cytosolic Nox2 co-factors) to the plasma membrane where the cytosolic proteins interact with the cell membrane-localized Nox2 complex (consisting of the proteins Nox2 plus p22^phox^) [4,6,7]; upon activation, the Nox2 protein can transfer an electron from NADPH to molecular O_2_ to generate O_2_**^−•^**. Dismutation of O_2_**^−•^**, either spontaneously or through the activity of superoxide dismutase, produces H_2_O_2_. For some time, these ROS have been known to play an important role in bodily defenses against infection through their bactericidal activity [2,8]. Recent studies have demonstrated that ROS also play an essential role in normal cell biology, providing the molecular signals that regulate cell motility, cell division, programmed cell death, and various other basal biological functions [9,10].

An outstanding question regarding the regulation of Nox2 relates to the upstream signals that initiate the Nox2 activation cascade. Previous studies have indicated that a phospholipase A_2_ (PLA_2_) enzyme plays an important role [11,12,13]. PLA_2_ represents a large family of lipid hydrolases that includes both secreted proteins that function extracellularly and intracellular proteins with specific subcellular localizations [14]. Furthermore, although all PLA_2_s hydrolyze phospholipids at their *sn*-2 position, many show some phospholipid substrate specificity. Earlier, the enzyme known as cytosolic PLA_2_ (cPLA_2_) was suggested as responsible for Nox2 activation [15], but that was not confirmed by a comprehensive study with cPLA_2_ null mice [16].

We recently have described the PLA_2_ activity of peroxiredoxin 6 (Prdx6) and shown that it is specifically involved in activation of Nox2 [7,17]. This PLA_2_, that we have called aiPLA_2_ based on its activity at acidic pH, is expressed widely in tissues. [18,19]. Prdx6 is present mainly in cytosol where it has minimal PLA_2_ activity but phosphorylation (via MAP kinases) results in its translocation to the plasma membrane where it binds to cell membrane phospholipids resulting in a marked increase in PLA_2_ activity [18]. The enzyme also is present in some organelles with acidic internal content such as lysosomes and lysosomal-related organelles; these organelles express aiPLA_2_ activity in the non-phosphorylated state, albeit at a considerably lower level than for the phosphorylated protein [18]. Nox2 activation by Prdx6 is prevented by the presence of MJ33, a mimic of the substrate transition state, that inhibits aiPLA_2_ activity [17,18,20].Activation of Nox2 also is prevented by mutating Prdx6 to a form (D140A-Prdx6) that specifically lacks a crucial member of the catalytic triad necessary for aiPLA_2_ activity.

Our recent studies have clarified the role of aiPLA_2_ in the activation of Nox2 [7,17]. Lipid binding of phosphorylated Prdx6 at the cell membrane is primarily to phosphatidylcholine and the PLA_2_ activity of Prdx6 with this substrate generates lysophosphatidylcholine (LPC) that, in turn, can be converted via lysophospholipase D activity to lysophosphatidic acid (LPA). LPA binds to its cell membrane receptor resulting in the generation of active Rac. The pathway for the role of aiPLA_2_ in Nox2 activation is shown in Figure 1. 

The impetus for our initial investigation of the interaction of Prdx6 with lung surfactant protein A (SP-A) was a publication showing that a serum protein from the Habu snake with homology to lung surfactant protein A (SP-A) inhibited the PLA_2_ component of its venom, presumably as a way to prevent accidental poisoning [21]. We confirmed that SP-A itself inhibits Habu snake PLA_2_ activity and showed further that SP-A can bind to Prdx6 and inhibit its aiPLA_2_ activity [22,23,24]. Prdx6 is present in lung lysosomal-related organelles, i.e., epithelial cell lamellar bodies (LB), the site of lung surfactant storage [25]. Since SP-A also is present in LB, we postulated that the interaction of Prdx6 with SP-A may be an important physiological mechanism to control the remodeling pathway for the synthesis of lung surfactant phospholipids [23]. Following these initial observations, we identified a 16 amino acid (aa) peptide sequence in SP-A that binds to Prdx6 and inhibits its aiPLA_2_ activity [26]. This peptide was identified with the ZDOCK program [27] by using molecular sequences in the Protein Data Base (PDB) in order to predict the molecular sites for interaction of the two proteins (Prdx6 and SP-A). Molecular modeling of SP-A used the sequence between the hydrophobic neck region and the collagen-like carbohydrate recognition domain (CRD) of rat/mouse SP-A [28]. Binding by the predicted SP-A-derived peptide to the Prdx6 protein was confirmed by isothermal titration calorimetry, circular dichroism measurement, and inhibition of its aiPLA_2_ activity [26]. The purpose of the present investigation was to determine the minimal amino acid sequence for an SP-A-derived inhibitory peptide and to evaluate the properties of inhibitory peptides as potential therapeutic agents to prevent Nox2 activation in conditions where excess ROS generation can lead to organ injury. This publication is part of a Forum on Peroxiredoxin 6 as a Unique Member of the Peroxiredoxin Family. 

## 2. Methods

Mice (C57Bl/6) were purchased from The Jackson Laboratory and maintained in the University of Pennsylvania Animal Care Facilities. We have described previously the generation of D140A-Prdx6 expressing mice using ‘knock-in’ technology; the lungs of these mice do not express aiPLA_2_ activity [29]. Subsequently, the neomycin resistance cassette (NRC) and the flippase gene that was introduced to remove the NRC have been removed from the genome of this mouse [19]. All studies using mice were approved by the University of Pennsylvania Institutional Animal Care and Use Committee (IACUC). 

Peptides were synthesized by APeptide Co., Ltd. (Shanghai, China) and were supplied with a mass spectroscopic analysis to verify composition; purity as determined by mass spectroscopy was 94% for PIP-1, 89% for PIP-2, and 91% for PIP-3.

For administration to mice, peptides were incorporated into liposomes that were prepared as previously described [30]. Liposome composition was dipalmitoylphosphatidylcholine (DPPC): egg phosphatidylcholine: phosphatidylglycerol: cholesterol in the molar ratio 50:25:10:15. To measure the in vivo effect of the peptides, mice were injected with PIP-2 by either the intratracheal (IT) or the intravenous (IV) route and were sacrificed at intervals; lungs were removed from mice and homogenized for measurement of aiPLA_2_ activity. 

The PLA_2_ activity of Prdx6 was measured as described previously [30]. The substrate was dipalmitoylphosphatidylcholine (DPPC) with ^3^H-9,10-palmitate in the *sn*-2 position of DPPC incorporated into lung surfactant-like liposomes as described above; substrate was incubated with enzyme for 60 min and activity was calculated from the liberation of ^3^H-palmitate as measured by scintillation counting.

The generation of ROS by the intact isolated perfused lung was determined from the increased fluorescence associated with the oxidation of Amplex red in the presence of horseradish peroxidase as described previously [17]. Aliquots of perfusate taken at intervals with increasing time of perfusion were analyzed with a spectrofluorometer. Angiotensin II (AngII) was added to lung perfusate to stimulate ROS production through its activation of N_OX_2, primarily in the lung endothelial cells since these cells express the relevant receptor.

## 3. Results

### 3.1. Minimal Effective Peptide Sequence

As indicated above, we previously identified a 16 amino acid peptide corresponding to the amino acids at positions 83 to 99 of the carbohydrate-recognition domain (CRD) of rat SP-A; these aa are located at positions 102–118 in the SP-A monomer. The corresponding sequence in the mouse SP-A CRD is identical in aa composition to the rat sequence. This 16-aa peptide inhibits the aiPLA_2_ activity of human recombinant Prdx6 by ~66% (Table 1), presumably by binding to Prdx6 as demonstrated previously [26]. We then synthesized peptides in which amino acids were deleted from either the N-terminus or the C-terminus of the 16-aa peptide in order to determine a minimum aa sequence that maintained the ability to inhibit aiPLA_2_ activity. Deletion of the C-terminal amino acid (Leu) abolished the ability of the peptide to inhibit aiPLA_2_ activity indicating that this aa is required for effectiveness (Table 1). Deletion from the N-terminus to generate 14-, 12-, 10-, or 9-aa peptides had no effect on the inhibitory activity of the peptide but activity was lost with the eight-aa peptide (Table 1). Thus, the minimal effective sequence for inhibition of aiPLA_2_ activity is a nine amino acid peptide comprising positions 8 through 16 of the original 16-aa peptide; this nine-aa peptide was called PIP-1.

We next identified the 16-aa peptide in human SPA that was analogous to the rat/mouse sequence and showed that it also inhibited the aiPLA_2_ activity of recombinant human Prdx6 (Table 2). Similarly to the rat/mouse peptide, deletion of seven aa from the N-terminus maintained its ability to inhibit aiPLA_2_ activity while deletion of the C-terminal aa abolished its inhibitory activity (Table 2). The active nine-aa peptide derived from the human SP-A sequence was called PIP-2. 

On comparing the PIP-1 (rat/mouse) and PIP-2 (human) peptides, the first aa and the terminal 4 aa were identical. Thus, only aa 2 to 5 differed between the two peptides. To evaluate the aa requirements at positions 2–5 for activity, we randomly substituted these with the aa that occurred in the effective sequences (PIPs-1 and -2). The nine-aa peptide LYDIRHQIL, representing a hybrid between the human and rat/mouse sequences, was an effective inhibitor of aiPLA_2_ activity (~70% inhibition, similar to PIP-2); this peptide, was called PIP-3. Of note, PIP-3, unlike PIPs-1 and -2, is not part of a known naturally occurring protein. Substitution of an I for the F at position 4 in the human peptide decreased the ability to inhibit aiPLA_2_ by about one-half. An additional 4 peptides synthesized with various combinations of the rat and mouse aa located at positions 2–5 of PIP-1 and PIP-2 were ineffective as aiPLA2 inhibitors (not shown). Finally, we synthesized six different peptides based on the PIP-2 sequence with substitutions at position 6 (Q for H), 7 (K,H,or R for Q), and 8 (V or A for I); none of these substituted peptides were effective inhibitors of aiPLA_2_ activity. 

### 3.2. Species Variability 

The differences between the human and rat/mouse sequences led us to examine the protein database and to extract the amino acid sequences of SP-A for various mammalian species, including five non-human primates, and one avian species (chicken) (Table 3). Only the L at position 9 in the peptide was invariant for these 19 species that were found in the database. An L occurred at position 1 in all but one species (guinea pig) and only one specie (chicken) did not have an H at position 6. The aa at positions 2–5 for the various species generally correspond to either the human or rat peptide sequence or to a combination of those amino acids. Species that show similar sequences to the rat/mouse model for aa in positions 2–5 include the cotton rat, cow, sheep, yak, and pig (Table 3). All sequences of non-human primate SP-A were identical to the human sequence as was the sequence for African elephant SP-A, although the latter was located at position 142–150 in the full length protein instead of position 111–119. The sequences from horse and wolf have a single difference (compared to human) for the aa at position 4. The consensus amino acids for each of the 9 positions were: (1) L,*F*; (2) H,Y,*L*; (3) D,E,*L,N*; (4) F,I,L*,N*; (5) R,K; (6) H,*Q*; (7) Q,K,*H,R*; (8) I,V,*A*; (9) L, where the amino acids in italics indicate an amino acid present in only a single species. The greatest differences from the human and rat/mouse sequences were seen for the guinea pig, rabbit, and chicken peptides.

Of note, the nine-aa peptide sequence of the wolf protein did not inhibit aiPLA_2_ activity while the sequence from the horse protein was only ~50% effective. These findings indicate that not all naturally occurring sequences are effective inhibitors (at least, of the human protein) and, if so, suggest that the inhibition of aiPLA_2_ by SP-A may not represent an important physiologic mechanism for regulating activity of this enzyme. 

### 3.3. Physical Properties of Inhibitory Peptides

The aa sequences and some characteristics of the three inhibitory peptides, PIP-1, PIP-2, and PIP- 3, are shown in Table 4. These peptides have a molecular mass of 1156–1178 with a predicted isoelectric point in the slightly basic range. Each of the three peptides contains four hydrophobic amino acid residues, with two of those on the same surface, and one negatively charged hydrophilic aa. The PIP-2 sequence has three while PIPs-1 and -3 have only two positively charged aa. PIP-1 is slightly more hydrophobic than the other two peptides but, in our experience, all three peptides readily dissolve in aqueous solution. The protein binding potential (Boman index) was slightly greater for the more hydrophilic PIP-2 indicating that it may bind slightly better to proteins as compared to PIPs-1 and -3. The UV extinction coefficient of PIPs 1 and 3 reflects the presence of a Tyr residue while PIP-2 does not contain an aa that absorbs UV light. No antigenic determinants were identified in these peptides as determined by the free on-line program provided by the Immunomedicine Group of the Universidad Complutense (Madrid, Spain). The chemical stability of PIP-2 was tested by placing the dry powder on the laboratory shelf at room temperature; its inhibition of the aiPLA_2_ activity of recombinant human Prdx6 was unchanged when measured after nine months of shelf storage indicating good stability of the peptide as a dry powder (data not shown); stability of the other peptides was not tested.

### 3.5. Inhibition of aiPLA_2_ Activity In Vitro

The effect of the PIP peptides on the aiPLA_2_ activity of human recombinant Prdx6 was determined by *in vitro* assay; the effective concentration of inhibitor was corrected for purity of the PIP preparation. Saturating concentrations of all three peptides (PIP-1, PIP-2, PIP-3) inhibited aiPLA_2_ activity by ~70% (Figure 2). The basis for the remaining 30% of activity that was not inhibited is unclear. The concentration of peptide (µg/mg Prdx6) for inhibition of aiPLA_2_ activity by 50% was ~8.3 for PIP-1, ~6.6 for PIP-2, and ~10 for PIP-3 indicating slightly greater effectiveness of PIP-2 as an inhibitor. The ratio of the molecular mass of the nine-aa peptides to Prdx6 (monomeric MW~25 kDa) is ~0.047 while maximal inhibition of activity was seen at a ratio of PIP to Prdx6 concentration of ~0.030 (Figure 2); this indicates a stoichiometry of approximately 1.5 to 1 for molar binding of the peptide to the Prdx6 monomer at maximal inhibition of activity. 

As determined by protein truncation and ITC, the site on Prdx6 for binding of the 16-aa peptide derived from rat/mouse protein was in the Prdx6 C-terminus (aa 210–225). This sequence does not include the aiPLA_2_ active site (S32-D140-H26) [33]. Thus, unlike MJ33 that competitively inhibits aiPLA_2_ activity as an analogue of the substrate transition state [34], inhibition of aiPLA_2_ activity by the SP-A derived peptide is non-competitive. The mechanism for inhibition of aiPLA_2_ activity through binding of the peptide to the enzyme is an alteration of the secondary structure of the protein, as shown by far UV circular dichroism [21]. 

### 3.6. Intracellular Delivery of PIP-2

Most small peptides are unable to cross cell membranes and thus their activity toward intracellular targets requires either aa modification or the use of a ‘delivery vehicle’. In order to evaluate the permeability of lung cells to PIP-2, we perfused isolated lungs for 15 min with the peptide alone or with the peptide encapsulated in liposomes; the perfusing medium was then switched to PIP-2 free (to remove any PIP-2 containing extracellular medium), and then Ang II was added to activate endothelial Nox2. Finally, Amplex red (plus horseradish peroxidase) was added to the recirculating perfusate in order to measure ROS production. Lung perfusion and ROS production were determined similarly at 15 min after intratracheal instillation of PIP-2, with or without liposomes. Our previous studies in rats and mice have shown that intratracheal instillation of these liposomes results in their rapid distribution throughout the lung and their rapid uptake by lung cells [35]. The linear rate of increase of oxidized Amplex red in the perfusate of control lungs (no PIP) indicates a continuous rate of Ang II-stimulated H_2_O_2_ production during the 1 h perfusion period (Figure 3). Perfusion with PIP-2 dispersed in saline (no liposomes) essentially had minimal effect on lung ROS production. Treatment of lungs with PIP-2 in liposomes, administered either by the IV or the IT route, led to a marked decrease in the rate of ROS production, compatible with the inhibition of lung aiPLA_2_ activity so that lung NOX2 was not activated by Ang II. PIP-2 incorporated into surfactant-like liposomes prior to lung perfusion resulted in ~75% inhibition of ROS production and results were similar for administration by either the IV or IT route (Figure 3). A similar experiment (not shown) with Nox2 null lungs showed a rate of Amplex red oxidation similar to the PIP-2 inhibited rate, indicating a low level of Amplex red oxidation by non-Nox2 dependent sources of ROS.

To confirm the requirement for liposomes as a delivery vehicle, lungs were perfused for 15 min with PIP-2 incorporated or not into liposomes, then perfused with fresh medium to clear extracellular PIP-2, and then homogenized and assayed for aiPLA_2_ activity. PIP-2 administered without liposomes had no effect on aiPLA_2_ activity of the lung homogenate while PIP-2 in liposomes inhibited activity by ~80% (Table 5). 

### 3.7. Effect of PIP-2 on aiPLA_2_ Activity In Vivo and its Biological Stability

To determine the effect and duration of a single dose of PIP-2 on aiPLA_2_ activity of mouse lungs *in vivo*, mice were injected with PIP-2 in liposomes by either the intratracheal (IT) or the intravenous (IV) route; individual lungs were removed at intervals, homogenized, and assayed for aiPLA_2_ activity. Lung aiPLA_2_ activity was inhibited by slightly greater than 80% after PIP-2 injection by either the IV or IT route (Figure 4). PLA_2_ activity in lungs remained depressed for 24 h, had begun recovery by 36 h, and was recovered fully by 72 h. The half-time for recovery was estimated at ~50 h. 

### 3.8. Specificity of PLA_2_ Inhibition

Lung cells, like those of other organs, have multiple PLA_2_ enzymes [14]. We evaluated the specificity of PIP-2 for inhibition of Prdx6 by measuring aiPLA_2_ activity in the lung homogenate of wild type and D140A-Prdx6 mutant mice. Lungs from the mutant mice do not express aiPLA_2_ activity since D140 is a key component of the aiPLA_2_ catalytic triad [29,33]. Measurement of PLA_2_ activity at pH 4 in the absence of Ca^2+^ is relatively specific for the PLA_2_ activity of Prdx6 (aiPLA_2_) [18]. This activity was inhibited by 91% following addition of PIP-2 to the assay of the lung homogenate (Table 6). Although Prdx6 has relatively little PLA_2_ activity when measured at pH 7, other lung PLA_2_ enzymes, some of which are Ca^2+^-dependent for activity, show activity at neutral pH; this activity was not affected by mutation of D140 in Prdx6 (Table 6). Unlike the effect of PIP-2 on aiPLA_2_ activity at pH 4, the addition of PIP-2 had no effect on activity of the lung PLA_2_ enzymes that are active at pH 7. Thus, aiPLA_2_ appears to be the only PLA_2_ enzyme, at least in the lung, that is inhibited by PIP-2. That result is not surprising since the specificity of binding of PIP-2 to the Prdx6 protein is the basis for its inhibition of aiPLA_2_ activity and other PLA_2_ enzymes do not share aa sequence homology with Prdx6.

## 4. Discussion

The goal of the present investigation was to follow up on our published observation that a 16-aa peptide derived from the rat SP-A sequence could bind to Prdx6 and inhibit its aiPLA_2_ activity [26]. We first verified, through search of the Protein Data Base (PDB), that the rat and mouse sequences in the relevant part of the SP-A carbohydrate recognition domain (CRD), were identical. We next verified the corresponding region in the human SP-A CRD and confirmed that this peptide also inhibited aiPLA_2_ activity, similar to the rat/mouse peptide. We found that progressive deletion of seven aa from the N-terminus of either the rat/mouse or the human 16aa sequences had no effect on the inhibitory activity of the peptides, while activity was lost with deletion of eight aa from the N-terminus or deletion of the C-terminal aa. Thus, a nine-aa peptide represents the minimal sequence that was effective as an aiPLA_2_ inhibitory peptide. 

Examination of the sequences of the two peptides (human and rat/mouse) showed identical aa in position 1 and positions 6–9 (LxxxxQHIL), while peptides 2–5 varied. We have named these PLA_2_-inhibitory peptides as PIP-1 (rat/mouse) and PIP-2 (human). Substitution of aa 2 and 4 from the rat/mouse sequence into the corresponding positions of the human sequence resulted in a hybrid peptide (called PIP-3) that also had aiPLA_2_ inhibitory activity. However, a hybrid peptide with the aa from positions 3 and 5 of the rat/mouse sequence inserted into the human sequence was not active while insertion of an I for F at position 4 of the human sequence decreased activity by ~50%. Thus, only some aa in positions 2–5 supported activity of the peptide. Furthermore, a review of the natural sequences from a variety of species in the PDB indicated that, while the human sequence was well represented and found intact in non-human primates and the African elephant, other species showed variable aa in eight of the nine positions and only the terminal leucine was present in all. Thus, it is not possible to accurately predict the peptide sequences that will be effective as inhibitors of aiPLA_2_ activity and further testing of individual constructs will be necessary in order to fully grasp the role of individual aa and their combination for aiPLA_2_ inhibition. Since there is only modest conservation of the inhibitory sequence in various species while at least one corresponding naturally occurring sequence (in wolf SP-A) does not inhibit aiPLA_2_ activity, we conclude that SP-A modulation of aiPLA_2_ activity is unlikely to represent an important physiological function.

PIP-2 was selected as the prototype peptide for further testing. This peptide effectively inhibited the aiPLA_2_ activity of recombinant Prdx6 in vitro. Calculation of the concentration of peptide required for maximal inhibition of activity indicated binding that was approximately to 1.5:1 of inhibitor to enzyme on a molar basis. PIP-2, administered to mice by either the IT or IV route, also inhibited the endogenous aiPLA_2_ activity in the homogenized lung; the inhibition by either route of administration was similar. Like many peptides [36], PIP-2 was not able to cross the cell membrane and its effectiveness as an inhibitor required its administration with a liposomal carrier. Surprisingly, the peptide within the lung cells appeared to be relatively stable, and inhibition of aiPLA_2_ activity was demonstrated with a half-time of ~50 h after peptide administration. Testing of lungs with genetic absence of aiPLA_2_ (D140A-Prdx6 mutation) or under assay conditions that do not support aiPLA_2_ activity (pH 7) provide evidence that aiPLA_2_ is the only lung PLA_2_ enzyme that is inhibited by the PIP-2 peptide.

The ability of PIP-2 to inhibit the activation of Nox2 in response to angiotensin II was demonstrated with the isolated perfused lung preparation using Amplex red in the perfusate as a trap for the generated ROS. This result was expected based on our previous studies demonstrating that aiPLA_2_ activity in lungs is required for activation of Nox2 [7,17], although Prdx6 also may be involved in the activation of Nox1 [37]. At any rate, our previous studies have shown that nearly all ROS production in response to angiotensin II is abolished with ‘knock-out’ of Nox2 and that the addition of an aiPLA_2_ inhibitor to Nox2 null lungs has little additional effect [7,17,20]. Thus, the effect of PIP-2 in angiotensin II-treated lungs appears to be predominantly due to the inhibition of Nox2 activation with essentially no contribution due to inhibition of Nox1.

## 5. Conclusions

The present studies indicate that the PIP peptides can effectively prevent ROS generation by the intact mouse lung. Based on results with mouse lungs, both the inhibitory dose of PIP-2 as well as its persistence within lung cells are in a reasonable range for potential therapeutic application to treat conditions with excess ROS generation subsequent to Nox2 activation. One potential therapeutic use of the PIP peptides might be to prevent inflammation-mediated tissue injury such as occurs with the Acute Lung Injury syndrome [21]. As PIP-2 is derived from a normal lung protein, it is expected to be relatively non-toxic by itself, although its possible antigenicity as well as the long term effects of the inhibition of aiPLA_2_ activity will require further evaluation.

## Figures and Tables

**Figure 1 antioxidants-07-00181-f001:**
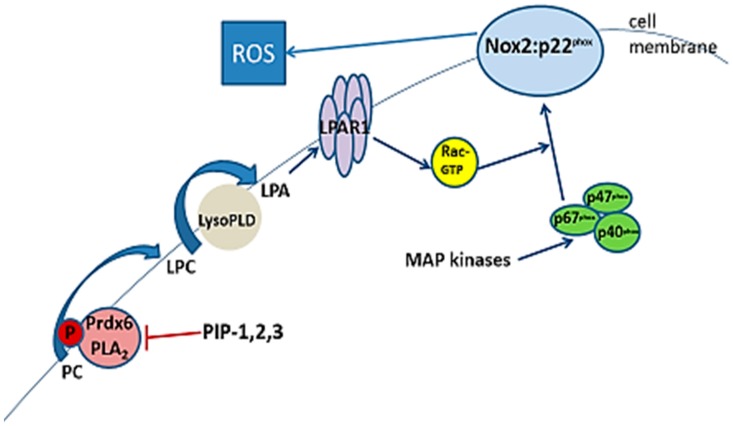
The pathway for activation of NADPH oxidase type 2 (Nox2). This pathway requires generation of lysophosphatidylcholine (LPC) through the PLA_2_ activity of phosphorylated Prdx6; LPC is converted to lysophosphatidic acid (LPA) that binds to its cell membrane-localized receptor (LPAR1) leading to the activation of Rac (1 or 2). Activated Rac (Rac-GTP) binds to p67^phox^ and it along with other phosphorylated cytoplasmic factors (shown in green) bind to the membrane-localized Nox2 complex resulting in its activation to generate reactive oxygen species (ROS). The presence of one of the PLA2-inhibitory proteins (PIP-1, -2, or -3) inhibits PLA_2_ activity and prevents Nox2 activation. PC, phosphatidylcholine; LysoPLD, lysophospholipase D; MAP kinase, mitogen activated protein kinase. Modified from [7] and reprinted with permission.

**Figure 2 antioxidants-07-00181-f002:**
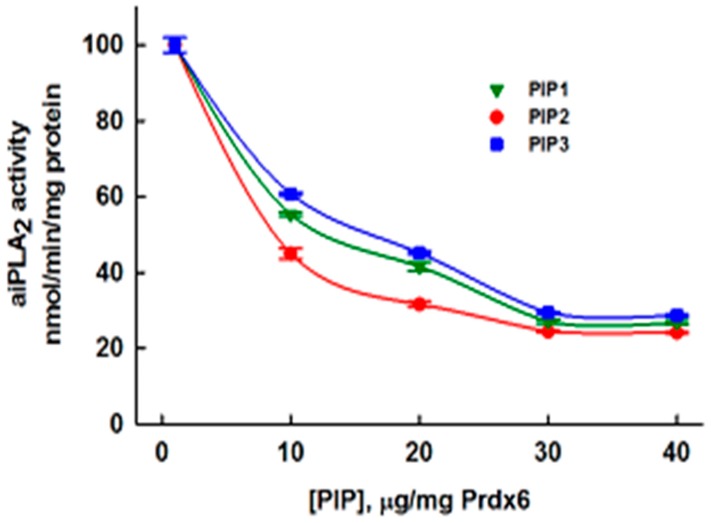
Effect of the concentration of PIP peptides on the aiPLA_2_ activity of recombinant human Prdx6. Results are mean + SEM for *n* = 3.

**Figure 3 antioxidants-07-00181-f003:**
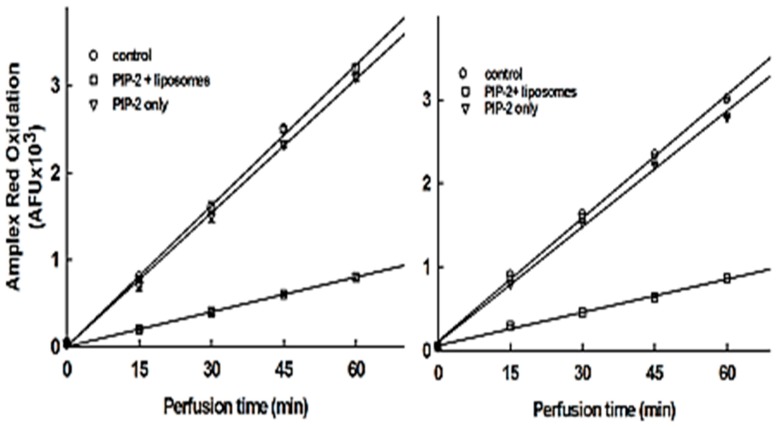
PIP-2 if encapsulated in liposomes inhibits lung ROS production when administered to. mice by either intratracheal (IT) or intravenous (IV) injection. PIP-2, with or without liposomes, and angiotensin II (to stimulate ROS production via NOX2) were administered to intact mice. Lungs were isolated 30 min later and ROS production was measured in the perfused lung by the oxidation of Amplex red plus horseradish peroxidase that were added to the lung perfusate.

**Figure 4 antioxidants-07-00181-f004:**
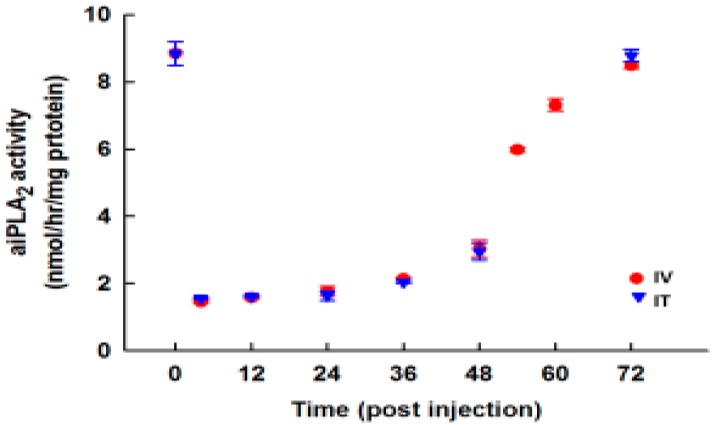
Persistence and stability of PIP-2 in vivo. PIP-2 was administered to mice either intratracheally (IT) or intravenously (IV) at 0 time. Lungs were harvested at intervals between 4 and 72 h after the PIP-2 administration, homogenized, and analyzed for aiPLA_2_ activity. Results are mean + SEM for *n* = 3–4.

**Table 1 antioxidants-07-00181-t001:** Deletion of N- or C-terminal amino acids from 16-aa rat-derived SP-A peptide: effect on the inhibition of the aiPLA_2_ activity of recombinant hPrdx6.

Peptide Length	Sequence Delete from N-term	Activity, nmol/min/mg Prot.	Sequence Delete from C-term	Activity, nmol/min/mg Prot.
control	No peptide	94.8	Scrambled peptide: LELDEEITEYQKQLHI	93.5
16 aa	DEELQTELYEIKHQIL	32.0	DEELQTELYEIKHQIL	32.0
14 aa	ELQTELYEIKHQIL	33.2	DEELQTELYEIKHQ	102
12 aa	QTELYEIKHQIL	31.5	No entry	
10 aa	ELYEIKHQIL	28.6	No entry	
9 aa	LYEIKHQIL (PIP-1)	32.3	ELYEIKHQI	89.6
8 aa	YEIKHQIL	94.4	DEELQTEL	93.6

Sequential deletions from the N-terminus or C-terminus of the rat/mouse 16-aa inhibitory peptide were used to determine the minimal effective sequence to inhibit the aiPLA_2_ activity of peroxiredoxin 6 (Prdx6). The results represent one assay done in duplicate.

**Table 2 antioxidants-07-00181-t002:** Deletion of amino acids from the N- or C-terminus of the 16-aa human-derived SP-A peptide: effect on the inhibition of aiPLA_2_ activity of recombinant hPrdx6.

Peptide, Number of Amino Acids	Sequence	Activity, nmol/min/mg Prot.	Comment
No peptide	Control	92.0	No added peptide
16 aa	DEELQATLHDFRHQIL	45.0	16 aa human peptide
10 aa	TLHDFRHQIL	31.5	Delete from N-term
9 aa	LHDFRHQIL (PIP-2)	29.9	Delete from N-term
9 aa	TLHDFRHQI	89.6	Delete from C-term

Same experiment as in Table 1 but deletions were from the human-derived 16-aa inhibitory peptide. The results represent one assay done in duplicate.

**Table 3 antioxidants-07-00181-t003:** Amino acids at the 9 amino acid positions in SP-A between 111-119 for various mammalian and 1 avian species.

Species	1	2	3	4	5	6	7	8	9
Human *	L	H	D	F	R	H	Q	I	L
Primates †	L	H	D	F	R	H	Q	I	L
Elephant (African) #	L	H	D	F	R	H	Q	I	L
Horse	L	H	D	I	R	H	Q	I	L
Wolf	L	H	D	L	R	H	Q	I	L
Rabbit	L	H	E	L	R	H	H	A	L
Chicken	L	L	N	L	R	Q	R	I	L
Rat	L	Y	E	I	K	H	Q	I	L
Mouse	L	Y	E	I	K	H	Q	I	L
Cotton rat	L	H	E	I	K	H	K	I	L
Cow	L	H	E	I	R	H	Q	V	L
Yak	L	H	E	I	R	H	Q	V	L
Sheep	L	H	E	I	R	H	Q	V	L
Pig	L	H	E	I	R	H	Q	I	L
Guinea pig	F	H	L	N	K	H	K	I	L

* Human surfactant protein A1. _†_ gorilla, orangutan, baboon, chimpanzee, rhesus monkey; # peptide is found at aa142–150. Red letters indicate those aa that are not present in the respective positions in either the human or the rat sequence.

**Table 4 antioxidants-07-00181-t004:** Properties of inhibitory peptides.

Properties	Rat/Mouse (PIP-1)	Human (PIP-2)	Hybrid (PIP-3)
Number of residues	9	9	9
Sequence	LYEIKHQIL	LHDFRHQIL	LYDIRHQIL
Molecular weight, g/mol	1156	1178	1170
Hydrophobic residues On same surface Grand average hydropathy	L,I,I,L I,I 0.1333	L,F,I,L F,I −0.3333	L,I,I,L I,I 0.0666
Charged amino acids: neg pos Iso-electric point, pH	E K,H 7.7	D R,H,H 8.0	D R,H 7.8
Protein-binding potential, kcal/mol	0.33	2.3	1.58
Extinction coeff. at 280 nm, M^−1^cm^−1^	1490	0	1490
Antigenic propensity, average	1.077	1.057	1.072
Antigenic determinants	none	none	none

Calculations and estimations were made by Innovagen’s peptide calculator [27] and the Antimicrobial Peptide Database (APD) predictor [31]; antigenic determinants were evaluated by a program from the immunomedicine group of the Universidad Complutense Madrid, Spain [32].

**Table 5 antioxidants-07-00181-t005:** Liposomes are required for inhibition of aiPLA_2_ activity by PIP-2 in isolated mouse lungs.

Conditions	aiPLA_2_ activity nmol/min/mg Prot.
No PIP	8.72 ± 0.16
PIP-2 in saline	8.50 ± 0.26
PIP-2 in liposomes	1.55 ± 0.11

Lungs were perfused with PIP-2 added directly to the perfusate or encapsulated in liposomes prior to addition. After 15 min, lungs were cleared of perfusate, homogenized and aiPLA_2_ activity was measured. Results are mean ± range for *n* = 2.

**Table 6 antioxidants-07-00181-t006:** PIP-2 specificity as an inhibitor of aiPLA_2_ as shown by the lack of effect on other PLA_2_ enzymes.

Conditions	PLA_2_ activity, nmol/min/mg Prot.			
	pH 4		pH 7 + Ca^2+^	
	WT	D140A-Prdx6	WT	D140A-Prdx6
No inhibitor	8.7 ± 0.16	0.2 ± 0.03	8.5 ± 0.26	8.3 ± 0.26
+PIP-2	1.6 ± 0.10	0.2 ± 0.06	8.5 + 0.30	8.5 ± 0.10

Lungs from wild type (WT) and D140A-Prdx6 ‘knock-in’ mice were cleared of blood and homogenized; PLA_2_ activity of the lung homogenate was measured at pH 4 in the absence of Ca^2+^ and at pH 7 in the presence of Ca^2+^. D140A-Prdx6 does not express aiPLA_2_ activity. Results are mean ± range for *n* = 2.

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
