# Peer review of "Identification of Small Peptides that Inhibit NADPH Oxidase (Nox2) Activation"

_antioxidants, 2018, doi:10.3390/antiox7120181_

Round 1
Reviewer 1 Report
COMMENTS TO AUTHORS
This is a well-conceived research and a well written research report by a group which contributed greatly to our present understanding of peroxiredoxin 6 (Prdx6), with special emphasis on its role in the regulation of the Nox2 NADPH oxidase system in general and in the pathogenesis of acute inflammatory lung diseases, in particular.
This report is an important follow-up on earlier work by the group on a peptide inhibitor, derived from lung surfactant protein A (SP-A), which binds to Prdx6 and inhibits its PLA2 activity and, consequently the activation of Nox2. The authors now determine the minimal effective sequence of the PLA2 inhibitory peptide in the rat/mouse (PIP1) and in humans (PIP2). The peptides were active in vitro on the PLA2 activity of Prdx6a and in vivo on the PLA2 activity and consequent Nox2 activation in the lung, following intratracheal or intravenous administration.
Here is a list of comments and queries addressed to the authors. These are listed in the order of their appearance in the text.
In the Abstract, placing Nox2 in brackets following the term “NADPH oxidase” is misleading since, as the authors mention in the paper, there are five families of Noxes.
Readers not familiar with the earlier work of Dr. Fisher’s group might require a brief “refresher course” on the mechanism by which the PLA2 activity of Prdx6 contributes to the generation of Rac. The canonical concept of the mechanism by which Rac contributes to Nox2 activation, at least in phagocytes, is that it dissociates from its complex with RhoGDI and, following conversion to Rac-GTP, binds to p67-phox and assists in the translocation of p67-phox to the membrane, where it binds to and activates Nox2.
In the Introduction (p. 2) it is stated that Nox2 in macrophages “is localized to intracellular phagolysosomes, and in endothelium where it is localized in the plasma membrane” (text in inverted commas is an ad litteram citation). To what endothelium do the authors refer? The sentence requires rewriting in a more comprehensible form.
Also in the Introduction, the authors use both the terms Nox2 and gp91-phox. This is confusing to the non-expert. I suggest that they stick to Nox2. It is less confusing to write that the Nox2 complex is a heterodimer of Nox2 and p22-phox.
The statement in the Introduction that generation of H2O2 from superoxide produced by Nox2 can occur via enzymatic dismutation should refer only to superoxide which leaked into the cytosol. H2O2 derived from superoxide released into the phagosome is the result of spontaneous nonenzymatic dismutation, not involving SOD.
This reviewer is not an expert in lung physiology and assumes that some of the readers might also fall in this category. It is, thus, recommended to make it clear in what cells in the lung does angiotensin II activate Nox2 to generate superoxide.
In tables 1 and 2 on p. 5, the number of experiments from which these results are derived should be indicated.
In the legend of Table 4, concentration of peptide/Prdx6 is given in ug/mg. Would not a mol/mol way of expression be more appropriate?
In the legend of Fig. 2, it is stated that results are “mean ± SEM”. However, in the graph proper, the error bars are not visible. Is this due to the fact that they are so small as to be covered by the symbols?
All the above comments relate to minor issues that do not modify the importance and the value of the manuscript.
Author Response
1) added type 2 after NADPH oxidase
2) added additional description of the role of Rac as suggested,along with an additional reference.
3) sentence re-written and additional info added re cell localization.
4) deleted GP91phox as suggested
5) the word 'spontaneous' was already in the sentence; added 'non-enzymatic'
6) the endothelial cells express the receptor for AngII--this has been added to the text
7) added that assays were done once in duplicate.
8) I believe that you mean Fig 2.Yes,mol/mol might be more appropriate,but we did the expt as
ug/mg.Either one can be calculated from the other--I would prefer to leave it as it is.
9) we have redrawn the figure to show the SEM
Thank you for making the effort to point these issues out to us.
Reviewer 2 Report
This is an excellent study, congratulations.
In this current manuscript, the authors continue the work of a previous study examining a short peptide shown to inhibit the phospholipase A2 activity of Prdx6. Using deletion studies, this current study was able to identify that a 9 amino acid portion of the original peptide is required for the aiPLA2 inhibitory effect. Investigation of sequence alignment between a number of species enabled the authors to identify a consensus sequence required for the active peptide.
In vivo experiments were carried out to examine the effect of the human peptide referred to as PIP-2. Delivery of PIP-2 via liposomes directly into the lungs of mice was able to inhibit Nox2 induced ROS production in response to angiotensin II. Based on these results, the authors suggest that this peptide represents a possible therapeutic agent in the prevention of lung inflammation.
The authors should be commended on their thorough experimental design and clear manuscript.
Author Response
Thanks for the review
Reviewer 3 Report
In a previous work the authors identified a 16 aa peptide from the rat surfactant protein A (SP-A) as an inhibitor of the Phospholipase A2 activity (aiPLA2) of Prdx6. In this manuscript the authors identify the minimal sequence as a 9 aa peptide. By comparison of the sequences in different mammals species and chicken they also identify the consensus sequences of the active peptide (LxxxxQHIL), and three active pepetides, one from the rat/mouse sequence (PIP-2), one from the human sequence (PIP-2), and a hybrid peptide (PIP-3).
The author convincingly show the ability of these three pepetides to inhibit aiPLA2 activity in vitro and in vivo. Moreover, when incorporated into liposomoses, by inhibiting the aiPLA2, PIP2 is capable of inhibiting Nox2 ROS production in vivo in response to angiotensin II. Therefore, the authors suggest the possible use of these peptides as therapeutics against lung inflammation.
The experiments are well designed and the results are very solid an interesting. Moreover, the authors’ results show the therapeutic potential of the aiPLA2-inhibiting peptides.
Author Response
Thanks for the review
Round 2
Reviewer 1 Report
Two very minor errors left:
On page 2, the authors write "complex (consisting of the proteins gp91phox plus p22phox)"; In their response, the authors claim that gp91phox was deleted. It seems that it was not. Please replace gp91phox by Nox2 = "consisting of Nox2 and p22phox".
On page 3, "Activated Rac (Rac-GDP)" should be replaced by "Activated Rac (Rac-GTP)".
Author Response
both changes made as suggested.Thank you.
